# The Crystallization Morphology and Conformational Changes of Polypropylene Random Copolymer Induced by a Novel β-Nucleating Agent

**DOI:** 10.3390/polym16060827

**Published:** 2024-03-16

**Authors:** Bo Wu, Xian Zheng, Yanwei Ren, Hailong Yu, Yubo Wang, Huanfeng Jiang

**Affiliations:** 1School of Chemistry and Chemical Engineering, South China University of Technology, 381 Wushan Road, Tianhe District, Guangzhou 510640, China; 6066233@163.com (B.W.); renyw@scut.edu.cn (Y.R.);; 2Guangdong Winner New Materials Technology Co., Ltd., Gaoming District, Foshan 528521, China; zxflying1218@163.com (X.Z.); winner_je@163.com (H.Y.)

**Keywords:** polypropylene random copolymer, rare earth, β-nucleating agent, crystallization, high-resolution FTIR spectroscopy, conformational change

## Abstract

The crystal morphology and conformational changes during crystallization of a polypropylene random copolymer (PPR) are the basis for understanding its crystallization process. In this work, novel rare-earth β-nucleating agent WBN-28 was directly added into PPR to induce β-crystallization. The results of differential scanning calorimetry (DSC) showed that it has an excellent β-crystal-induced effect. The β-crystal content could surpass 85%, calculated from wide-angle X-ray diffraction (WAXD) data. The morphology of the β-crystal and α-crystal was intuitively observed via a polarizing optical microscope (POM). The β-crystallites were interconnected to naturally develop plate-like crystalline regions possessing a certain size, and the α-crystallites with sufficient thicknesses possessed a cross-hatched phenomenon. The bundle-like supramolecular structure of the β-crystal induced by WBN-28 was further observed via a scanning electron microscope (SEM). The conformational changes in the crystallization process of PPR were resolved via high-resolution infrared spectroscopy to understand its β-crystallization in depth. The conformational changes during the crystallization of PPR were found to be different from those of the isotactic polypropylene homopolymer (PPH); they had their own characteristics. This will provide guidance for understanding the β-crystallization of PPR in depth.

## 1. Introduction

PPR is an important copolymer of polypropylene. In its molecular chain, ethylene copolymerization units are randomly inserted into the propylene sequence. Since the random insertion of an ethylene copolymerization unit destroys the crystallization of the propylene sequence, the thickness and crystallinity of PPR decreases significantly [1]. Compared with PPH, PPR has higher toughness, higher ductility, moderate strength, and excellent transparency [2,3], which makes it widely used in all walks of life [4,5]. PPR needs to be processed from a molten state, and undergoes a crystallization process to become an industrial product. The crystallization process determines the arrangement of the polymer molecular chains and controls the composition of the microstructure. The type and morphology of crystal significantly influences the properties of the final product [6,7,8,9,10]. Therefore, study of the crystallization process is extremely significant.

The β-crystal of polypropylene has a great influence on the properties of products because of its unique structure. Shear yield and deformation of loose β-crystals can consume a lot of impact energy [11]. Therefore, the material has a higher toughness, especially low temperature toughness, without too much of a decrease in rigidity [12]. On the other hand, due to the slow mobility of the amorphous phase, the diffusion coefficient and solubility coefficient are lower, showing higher oxidation stability than α-PP [13]. The higher content of β-crystal means that the above properties are improved greatly, and manufacturers are always trying to obtain a higher content of β-crystal as much as possible [14]. The β-crystal of polypropylene is thermodynamically quasi-stable and kinetically unfavorable to generate. A variety of supramolecular structures may be assembled during the crystallization process, including spherulite [15], transcrystalline [16], cylindrite [17], shish-kebab [18], and hexagonal crystal [19]. The performance of the product is strongly affected by the supramolecular structure and morphology [20]. The crystalline morphology and microstructure of polypropylene can be clearly observed by using a polarizing optical and electron microscopic in a systematic manner [21]. This provides an effective means of better understanding crystallization and crystal growth in polypropylene. The introduction of an effective β-nucleating agent is the most reliable and prominent method for preparation of β-PP [22]. At present, there are many varieties of β-nucleating agents for polypropylene, including carboxylates [23], amides [24,25,26], inorganic salts [27,28], rare earths [29,30] and even polymers [31,32]. However, not all β-nucleating agents have an excellent β-induced nucleation effect for PPR [33]. The β-induced effect is often inhibited by the presence of copolymerized ethylene units or the coexistence of α, β, or γ-crystals in PPR [34,35,36]. The same nucleating agent cannot achieve the same nucleation effect in PPR as in PPH, and the β-induced effect in PPR is ordinarily worse than that in PPH [37]. At present, there are few β-nucleating agents that can be directly applied into PPR to induce large amounts of β-crystals [38]. Correspondingly, the number of studies on the morphology and supramolecular structure of the β-crystal in PPR are significantly less than those on PPH.

To better understand the crystallization process in depth, studying changes in the crystalline conformation are equally important, except for the observation of the crystalline morphology [39,40]. A few absorption bands are extremely sensitive to the physical state of the sample in the infrared spectra of crystalline polymers [41,42,43]. Depending on the source, these sensitive bands can be divided into two distinct categories. One is caused by intermolecular forces in the lattice where the polymer molecules are arranged in an ordered three-dimensional arrangement. These bands are often referred to as crystallinity bands. The other is associated with intramolecular vibrational coupling within a single chain, and these bands are defined as regularity bands or helix bands [44]. Most of the absorption bands of polypropylene below 1400 cm^−1^ belong to the regularity bands [45]. The specific regularity bands are related to the different critical lengths “n” of the molecular chain isotactic sequence. The n value is the corresponding number of monomers in helices and the different infrared bands represent different ordered helical lengths, shown in Table 1 [46]. The regularity band is a marker for tracking the gradual evolution of intramolecular order in polymer chains in the isotropic melt [47,48,49]. Consequently, information on conformational changes both before and during crystallization can be extracted by examining the evolution of the intensity of the key helical sequences in crystallization [50]. In isotactic polypropylene, the evolution from the melt to the crystalline state is usually investigated. The n values for appearance of bands at 841, 973, 998, and 1220 cm^−1^ are 12, 3~4, 10, and over 14 monomers in helical sequences, respectively [46,51]. The preceding analyses are based on isotactic PPH, while there are almost no studies on PPR.

In the present work, we introduced a novel rare-earth β-nucleating agent into PPR to induce β-crystals. The DSC and WAXD results showed that excellent β-crystal induction was achieved. The morphology of the β-crystal and α-crystal was intuitively observed via POM. Moreover, the supramolecular structure of the β-crystal induced by WBN-28 was observed via SEM. Finally, the conformational changes in the crystallization process of PPR were resolved via high-resolution infrared spectroscopy to understand its β-crystallization in depth.

## 2. Materials and Methods

### 2.1. Materials

The commercially available PPR (PA14D) used in this work was supplied by PetroChina Daqing Petrochemical (Daqing, China). Its melt flow rate was 0.42 g/10 min (230 °C, 2.16 kg), and its weight-average molecular weight M_w_ was 4.81 × 10^5^ g/mol. The ethylene content of PA14D was about 4.1%. The β-nucleating agent was a novel rare-earth (called WBN-28, an organic complex of barium and lanthanum with some specific ligand) kindly supplied by Guangdong Winner New Materials Co., Ltd. (Foshan, China).

### 2.2. Samples Preparation

A corotating twin-screw extruder (TSJ-35, L/D ration = 40, manufactured by Nanjing Norda Machinergy Equipment Co., Ltd., Nanjing, China) was used to prepare the samples containing different WBN-28 concentrations. For more convenience, the samples were abbreviated as PPR0, PPR0.05, PPR0.1, and PPR0.2, respectively, where the number represents the percentage of β-nucleating agent. The processing temperature was set to 165–205 °C from hopper to die, and the screw speed was fixed at 300 rpm. The extruded strands were immediately quenched in water and then cut into pellets by a pelletizer.

### 2.3. Characterizations and Measurements

#### 2.3.1. Differential Scanning Calorimetry (DSC)

The samples were in the weight range of 6–9 mg. They were measured via a Perkin Elmer Jade DSC (Waltham, MA, USA) using nitrogen as the purging gas. The heating rate was 10 °C/min in the first heating scan from 60 °C to 200 °C, then the samples were kept at 200 °C for 5 min to eliminate the thermal history. The cooling rate was 10 °C/min in the cooling scan from 200 °C to 60 °C. The subsequent heating scanning rate was 10 °C/min from 60 °C to 200 °C, and the melting endothermic peaks were recorded.

#### 2.3.2. Wide-Angle X-ray Diffraction (WAXD)

The samples were first held at 200 °C for 5 min. Subsequently, they were cooled to 120 °C and kept at a constant temperature for 2 h. In the process, the samples were pressed into thin sheets about 0.15~0.2 mm. After that, the sheets were cooled to room temperature. The 2D-WAXD experiments were carried out with a D8 ADVANCE (Bruker, Billerica, MA, USA) diffractometer using Cu Kα radiation scanned from 5° to 60° at a speed of 6 °/min. The wavelength of the X-ray was 0.154 nm at 40 kV and 40 mA. The data were processed by XRD pattern processing and identification (JADE 6.5). The relative amount of β-crystals was calculated according to Tuner-Jones equation [52]:(1)$$K_{\beta }=\frac{H_{\beta }(300)}{H_{\beta }\left(300\right)+H_{\alpha 1}\left(110\right)+H_{\alpha 2}\left(040\right)+H_{\alpha 3}(130)}*100\%$$
in which *H_β_*(300) is the intensity of (300) reflection of the β-crystal; and *H_α_*_1_(110), *H_α_*_2_(040), and *H_α_*_3_(130) are the intensities of the (110), (040), and (130) reflections of the α-crystal, respectively.

#### 2.3.3. Polarizing Optical Microscopy (POM)

The isothermal crystallization morphologies of the samples were observed via a Leica DM2700M polarizing optical microscope (Wetzlar, Germany) with a heating stage. The samples were heated to 200 °C for 5 min and pressed into thin sheets using a cover-slip. After that, the samples were cooled to 120 °C at a rate of 100 °C/min on a heating stage to avoid non-isothermal crystallization as much as possible.

#### 2.3.4. Scanning Electron Microscopy (SEM)

All samples were first isothermally crystallized at 120 °C for 2 h. Subsequently, they were cryogenically fractured in liquid nitrogen, and the cryo-fractured surface was immerged in a potassium permanganate solution [53] to etch the amorphous part of PPR in order to clearly observe the crystal structure. All samples were sputter-coated with gold powder for 240 S before the test. SEM experiments were carried out with a SU8220 (Hitachi, Tokyo, Japan) instrument.

#### 2.3.5. High-Resolution In Situ FTIR Spectroscopy

The samples were suppressed into a diaphragm of approximately 20 μm. Subsequently, they were held at 200 °C for 3 min on a Linkam TST350 heating stage (Salfords, UK). Crystallization was carried out by natural cooling from 200 °C, and the infrared signals during the crystallization process were collected via an IS50R (Thermofisher Scientific, Waltham, MA, USA) IR spectrometer. The MCT detector was selected for infrared signal collection, with a collection interval of 11 s, a collection resolution of 0.25 cm^−1^, and collection time of 4. Normalization of infrared absorption peak areas was carried out using infrared software (Omnic 9.0).

## 3. Results and Discussion

### 3.1. Measurement of β-Crystal

DSC and WAXD are frequently used to measure the content of β-crystals and characterize β-crystals [54]. There are obvious fusion peaks of α-crystal and β-crystal on the DSC heating curves, as shown in Figure 1a. The melting points typically correspond to 130.1 °C and 145 °C, respectively, in PPR0.05. The melting point increases slowly with the increment of the nucleating agent. As shown in Figure 1b, 2θ = 14.2°, 17.1°, 18.9°, 21.5°, and 22.2° typically correspond to the crystal planes (110), (040), (130), (111), and (−131), respectively, of α-crystal reflections in the WAXD pattern of PPR0. Moreover, a small amount of γ-crystal is generated during the isothermal crystallization process, and 2θ = 20.3° corresponds to the γ-crystal plane (117) [55]. An extremely strong diffraction peak of β-crystal plane (300) appears near 2θ = 16.3° after adding the β-nucleating agent. The α-crystal plane (110) and (040) diffraction peaks are gradually weakened with the increment of the nucleating agent. All samples essentially have no diffraction peaks of α-crystal plane (130), (−131), and γ-crystal plane (117) except for PPR0. The relative content of β-crystal calculated from WAXD data is 85.1%, 86%, and 86.5% in PPR0.05, PPR0.1, and PPR0.2, respectively. Test data from DSC and WAXD show that WBN-28 undoubtedly has an efficient β-crystal induction effect, even at low concentrations.

### 3.2. Observation of β-Crystal Morphology

The growth process of polypropylene spherulites and the gradual changes of crystal morphology can be observed visually using a polarizing microscope. Due to the unlike birefringence phenomena of α-crystals and β-crystals, they display distinctive optical images. It is therefore possible to clearly distinguish between the α-crystal and β-crystal of polypropylene and to accurately observe the crystallization or melting process [56,57]. Figure 2 shows the gradual evolution of the isothermal crystallization morphology of PPR0 with time at 120 °C. The picture is virtually completely dark when the crystallization starts. The matrix is amorphous except for a few nuclei formed by homogeneous nucleation. As shown in Figure 2c, a few crossed crystallites progressively appear in the matrix and slowly begin to grow with the passage of time, and the dark amorphous region gradually decreases. The volume of crystal increases further with the increment of crystallization time, collides with each other to naturally fill the whole image, and finally forms a larger α spherulite. There are still some amorphous dark areas in the picture when the crystallization time reaches 4 min, sufficiently indicating that the system crystallizes slowly. There are α spherulites of uneven size in the image when the crystallization is ultimately completed. In addition, the cross-hatched phenomenon of α spherulites [58] and the interface is relatively vague due to the existence of a rubber phase in PPR.

The crystallization is significantly accelerated after the addition of WBN-28 as shown in Figure 3. There are many slight nuclei in the image when the crystallization time is 1 min. The α spherulites do not continue to grow with crystallization time. However, β-crystallites grow rapidly and instantly connect with each other to form some plate-like regions, and obvious dark grey band boundaries can be observed. Compared with PPR0, the crystallization rate is significantly accelerated, and the grain size is refined even though the content of nucleating agent is merely 0.05%. As the crystallization time is further prolonged, the β-crystal regions become brighter, which proves that more β lamellae [59] are superimposed. As shown in Figure 3d–f, there is no significant change in the size of the crystalline regions, and the position of dark grey band demarcations is relatively fixed. It sufficiently shows that demarcation is not caused by the distortion of β lamella [11], but rather by the rubber phase in PPR. In addition, the α-crossed nuclei formed at the initial stage of crystallization do not grow continuously, but are randomly distributed in the β-crystal regions. The size of α spherulite in the visible image is smaller, and the cross-hatched phenomenon is weaker than PPR0.

As shown in Figure 4, the crystallization rate further increases with a larger increment of β-nucleating agent. Different from PPR0 and PPR0.05, a large number of β nuclei naturally appear in the image of PPR0.2 at the beginning of isothermal crystallization at 120 °C. The dark amorphous region is not observed in the image when the crystallization time reaches 1 min. Similar to Figure 3c, β-crystallites grow rapidly and instantly connect with each other to form plate-like regions, and obvious dark grey band boundaries can be observed. The most notable difference is that there is no α-crossed spherulite in the visible image. Due to the efficient β induction effect of WBN-28, enough β nuclei are naturally produced in PPR0.2. The rapid expansion of β-crystals inhibits the continuous growth of the α-crystal, and the latter can merely exist in the matrix with a limited number of nuclei. Because there is almost no coherent superposition of lamellae, cross-hatched spherulites are unobserved. As shown in Figure 4b–f, the crystalline regions become more brilliant, but the size does not change significantly. The dark grey band boundary exists at the same location, indicating that the β lamellae grow continuously at a fixed position with the crystallization process.

The crystal morphology of samples melted at 138 °C after crystallization is recorded in Figure 5. Since the α- and β-crystal of PPR naturally have different melting points, the isothermally crystallized samples are directly reheated to 138 °C to melt the β-crystal while retaining the α-crystal. The α-crystallites and their distribution in each sample can be directly observed to further understand the crystallization process. Noticeable change is unobserved from Figure 5a. The obvious cross-hatched phenomenon indicates that the α lamellae are stacked thicker, and that there are a large number of α spherulites. In Figure 5b, the grain size of the α-crystal is finer than that of PPR0, and the crossed polarization effect is obviously not as significant as that of PPR0. Moreover, the stacking thickness of α lamellae is not enough to produce a noticeable optical effect. Only the small crossed polarization effect of the α-crystal can be carefully observed. Simultaneously, the surrounding dark regions undoubtedly contain the molecular chain segments melted from the β-crystal and the rubber amorphous phase. The blurred image in Figure 5c sufficiently indicates that only a few unmelted α lamellae are present. The crossed polarization effect of the α-crystal is unobserved because the α lamellae are not stacked to a sufficient thickness. It further indicates that when a sufficient amount of WBN-28 is added into PPR, it will induce a large number of β nuclei. This makes β-crystals occupy an absolutely dominant position in the competitive growth with α-crystals, thereby inhibiting the growth of α-crystals.

Figure 6a is SEM photograph of PPR0. In Figure 6a, the bundled β-crystallites cannot be observed because there are only α-crystallites. The β-crystal supramolecular structure formed by the addition of WBN-28 into PPR can be clearly observed from Figure 6b,c. The SEM morphology of the two crystallite forms has been confirmed in a previous study [60]. A large number of bundle-like β-crystallites are present in the SEM images. The β lamellae are continuously stacked around the bundle center to naturally form β single spherulite, and the spherulites are compactly stacked. In this process, it is easy to collide with other spherulites that are rapidly growing at the same time, and then connect to develop a plate-like crystalline region. This is consistent with the direct POM observations. In addition, bundle-like spherulite exhibits higher creep resistance and toughness than radial spherulite [61], which also suggests that an appropriate amount of WBN-28 can significantly improve the performance of PPR.

### 3.3. Analysis of High-Resolution In Situ FTIR Spectroscopy

The molecular composition and crystal structure of WBN-28 are unknown. It is difficult to study specific crystallization process through lattice matching and interaction energy between the nucleating agent and polypropylene molecule. We attempted to investigate the conformational changes of PPR crystallization process via in situ infrared spectroscopy to understand the β-crystallization in depth.

Figure 7 is the infrared absorbance spectra of the molten and crystalline states of PPR0. The regularity bands below 1400 cm^−1^ are highlighted in the figure. Among the identified characteristic absorption peaks, only the ordered short chain segments of the shorter (n = 3~4) sequence (973 cm^−1^) monomer units exist in the molten PPR. The rest of the regularity bands labelled in red disappear, and 1153 cm^−1^ is independent of the ordered short-chain segments of the helical chain. The visible result is inconsistent with isotactic PPH. The long (n = 12 and n = 14) helical sequences (841 and 1220 cm^−1^) will melt and disappear, and the shorter (n = 3~4 and n = 10) sequences (973 and 998 cm^−1^) will persist in the molten isotactic PPH [51]. Because there is α-crystal in PPR0, it shows that shorter (n = 3~4) sequences (973 cm^−1^) play an important role in the formation of α-crystal.

Infrared spectroscopy is sensitive to the molecular helical conformation. The continuous infrared absorption spectra of PPR samples with various contents of nucleating agent during the cooling process are shown precisely in Figure 8a–c. The content and sequence of appearance of each regularity band in PPR reflect the gradual process of chain segment ordering during crystallization. Some regularity bands, such as those at 808, 973, and 1167 cm^−1^, are present in the molten samples. The corresponding number of monomers in the regularity bands of 808, 973, and 1167 cm^−1^ are ~7, 3~4, and ~6, respectively. This indicates that some short helical conformations still exist in the melt. Other regularity bands, such as 841, 900, 998, 1220, and 1303 cm^−1^, are absent, and they are all longer helical conformations. The regularity bands corresponding to the longer helical conformations emerge successively and increase concomitantly in intensity, naturally accompanying crystallization. There is no new regularity band in PPR accompanying the nucleating agent introduction. However, the representative canonical bands are shifted, as shown in Figure 8d. For polypropylene, the absorption peak at 840 cm^−1^ accurately reflects the crystalline shape and the degree of disorder of the α-crystal [62]. The absorption peak shifted gradually from 840.8 cm^−1^ to 841.3 cm^−1^ after gradual introduction of nucleating agent. It indicates that most of the molecular chains crystallize in the form of a β-crystal. Simultaneously, the degree of α-crystal is unperfect and its concentration is low. Abundant β-crystals and few α-crystals coexist in PPR0.2. This is consistent with the direct POM results for α-crystal, which can merely be observed as fine or unpolarized after melting β-crystallites.

Figure 9 shows the variation curves of the normalized area with cooling time. They are plotted from intensity variations of the characteristic absorption peaks, representing different helical ordered length chain segments below 1400 cm^−1^ during crystallization in Figure 8a–c. When the temperature is above 138 °C, although the nucleating agent is present, the growth rate of β-crystal is extremely low, and the molecular chain movement ability is too strong to crystallize. The acceleration effect of the nucleating agent on crystallization is not obvious. This is close to the 140 °C ceiling temperature of the optimum growth rate of β-crystal described by Varga [63]. Signal enhancement of the regularity bands of long helical conformations occurs below 138 °C. In the whole cooling crystallization process, the order of the area growth of several regularity bands is 840 > 900 > 998 > 1220 > 940 cm^−1^. The corresponding number of monomers in the regularity bands of 840 cm^−1^ is 12. This indicates the helical segments of 12 monomers are preferentially formed, then the longer ordered segments of 14 or more are gradually formed during crystallization. This is consistent with earlier calculations of critical length by Yan et al. [40]. Parallel ordering of chain segments for crystallization is induced only when the length exceeds a critical value in PPR.

Furthermore, the absorption peaks at 973 cm^−1^ appear to increase and then decrease, with the temperature dropping as shown in Figure 9a,b. Combined with direct POM observations of crystalline morphology, this suggests that α crystallization requires the formation of short ordered chain segments of 3–4 monomers as a crystallization transition. The β nuclei are insufficient to fully constrain a sequence of 12 monomers to crystallize in PPR0.05, so some α nuclei formed from short sequences are present and ultimately develop into α-crystallites. On the other hand, the same band rarely changes with decreasing temperature in Figure 9c. A large number of β nuclei are present in PPR0.2; this makes β-crystals occupy an absolutely dominant position in the competitive growth with α-crystals, thereby inhibiting the growth of α-crystals. Transition of short ordered chain segments is no longer required, and the nucleating agent WBN-28 can constrain longer ordered chain segments to form β-crystallites directly.

## 4. Conclusions

In this study, we report on novel β-nucleating agent WBN-28, which can be directly added into PPR to induce β-crystals. The relative content of β-crystal calculated from WAXD data is 86.5% in PPR0.2. Test data from DSC and WAXD show that WBN-28 undoubtedly has an efficient β-crystal induction effects, even at low concentrations of 0.05%. The addition of nucleating agents significantly improves the crystallization rate. Dark amorphous regions were unobserved via POM in PPR0.2 for less time than in PPR0.05 and PPR0. The β-crystallites grow rapidly and instantly connect with each other to form some plate-like regions, and obvious dark grey band boundaries can be observed. The crystalline regions become more brilliant, but the size does not change significantly with increasing crystallization time. The growth of α-crystallites is inhibited by β-crystallites, causing α crossed spherulite to be absent in the visible POM image of PPR0.2. The β-crystal supramolecular structure formed by the addition of WBN-28 into PPR is bundle-like, observed via SEM. The β lamellae are continuously stacked around the bundle center to naturally form β single spherulite, and the spherulites are compactly stacked. Gradual changes in the ordered structure occur during the crystallization process. Unlike isotactic PPH, only a shorter sequence (n = 3~4) of a 973 cm^−1^ band is present in molten PPR. It can serve as a crystallization transition to further constrain the longer sequences and naturally generate α-crystals. In PPR0.2, WBN-28 can constrain longer ordered chain segments of 12 monomers to form β-crystals directly, without the need for short sequences of 3–4 monomers as crystallization transitions. This study is beneficial to understand β-crystallization process in PPR in depth and provide further guidance for β-modification of PPR.

## Figures and Tables

**Figure 1 polymers-16-00827-f001:**
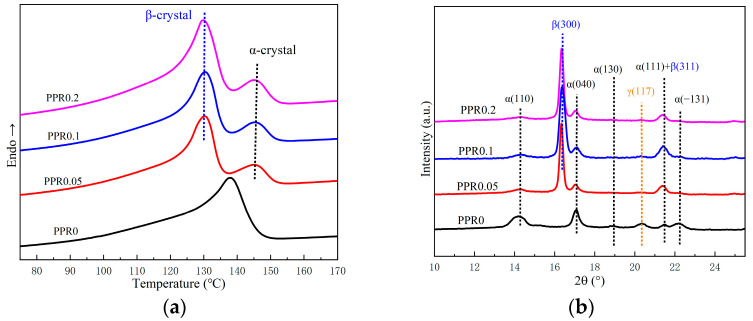
Test curves of PPR with various amounts of β-nucleating agent: (**a**) DSC curves; (**b**) WAXD curves.

**Figure 2 polymers-16-00827-f002:**
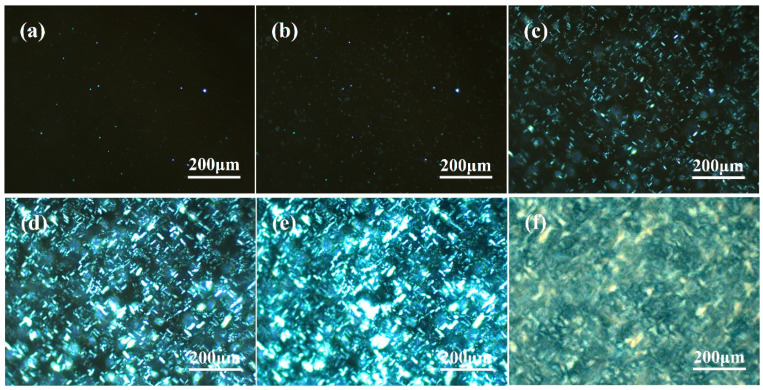
Isothermal crystallization morphology of PPR0 at 120 °C: (**a**) 0 min; (**b**) 1 min; (**c**) 2 min; (**d**) 3 min; (**e**) 4 min; (**f**) crystallization completed.

**Figure 3 polymers-16-00827-f003:**
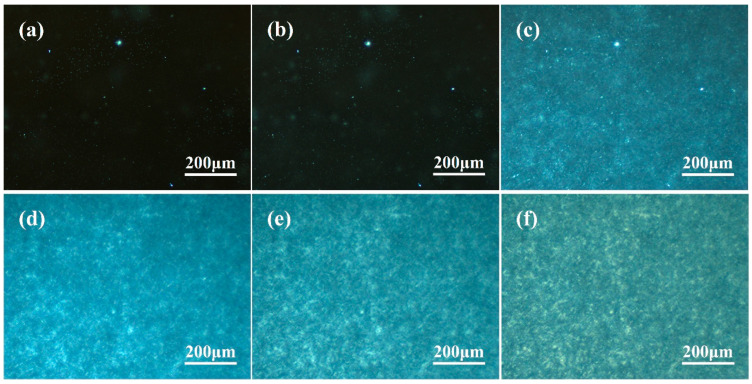
Isothermal crystallization morphology of PPR0.05 at 120 °C: (**a**) 0 min; (**b**) 1 min; (**c**) 2 min; (**d**) 3 min; (**e**) 4 min; (**f**) crystallization completed.

**Figure 4 polymers-16-00827-f004:**
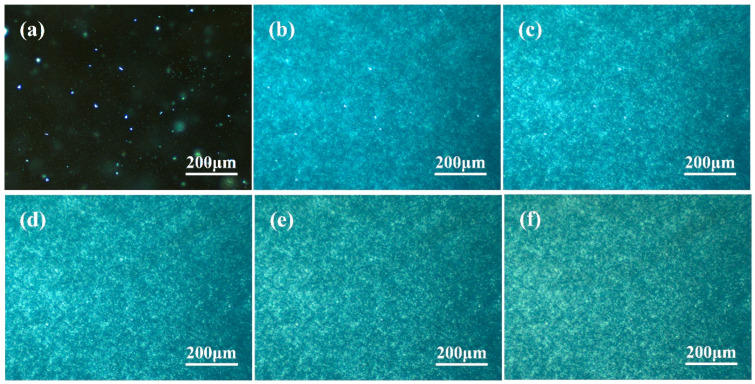
Isothermal crystallization morphology of PPR0.2 at 120 °C: (**a**) 0 min; (**b**) 1 min; (**c**) 2 min; (**d**) 3 min; (**e**) 4 min; (**f**) crystallization completed.

**Figure 5 polymers-16-00827-f005:**
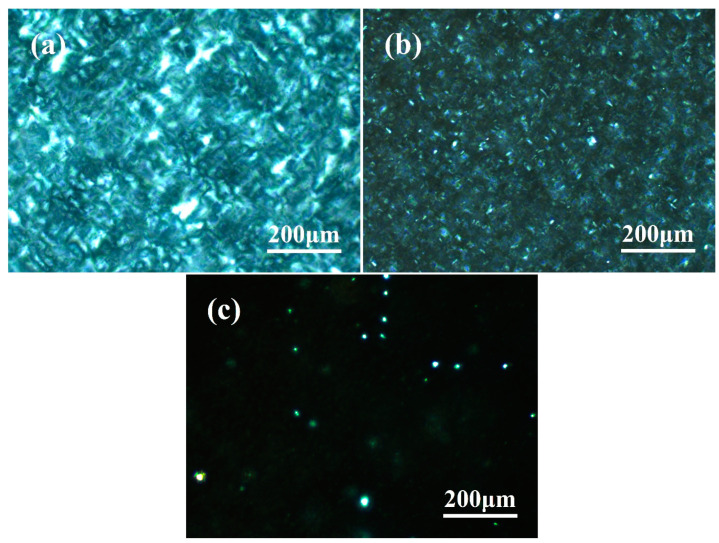
The crystal morphology of samples melted at 138 °C after crystallization: (**a**) PPR0; (**b**) PPR0.05; (**c**) PPR0.2.

**Figure 6 polymers-16-00827-f006:**
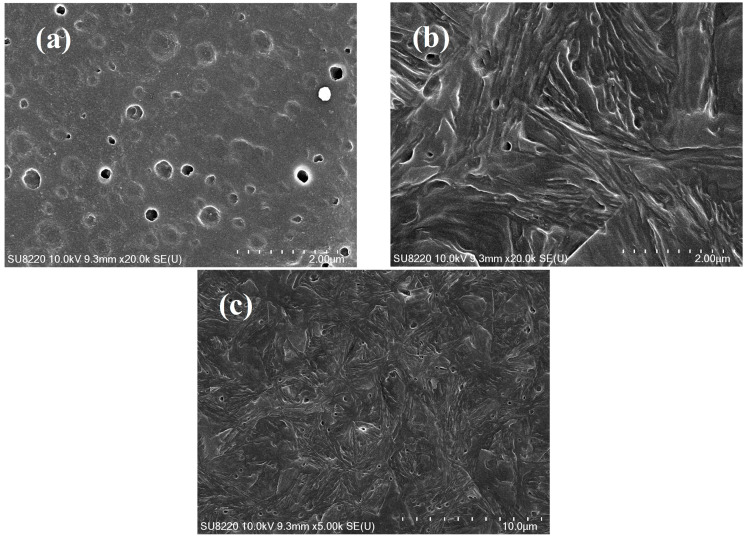
(**a**). SEM photograph of PPR0. (**b**,**c**) are the SEM photographs of PPR0.2 after isothermal crystallization at 120 °C: (**b**) ×25,000; (**c**) ×5000.

**Figure 7 polymers-16-00827-f007:**
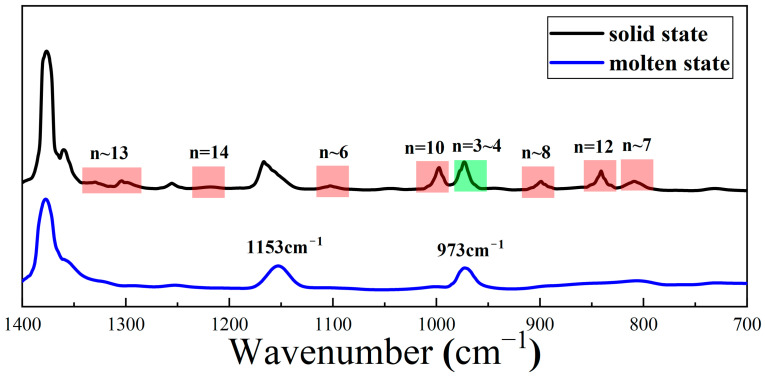
The infrared absorbance spectra of the molten and crystalline states of PPR0.

**Figure 8 polymers-16-00827-f008:**
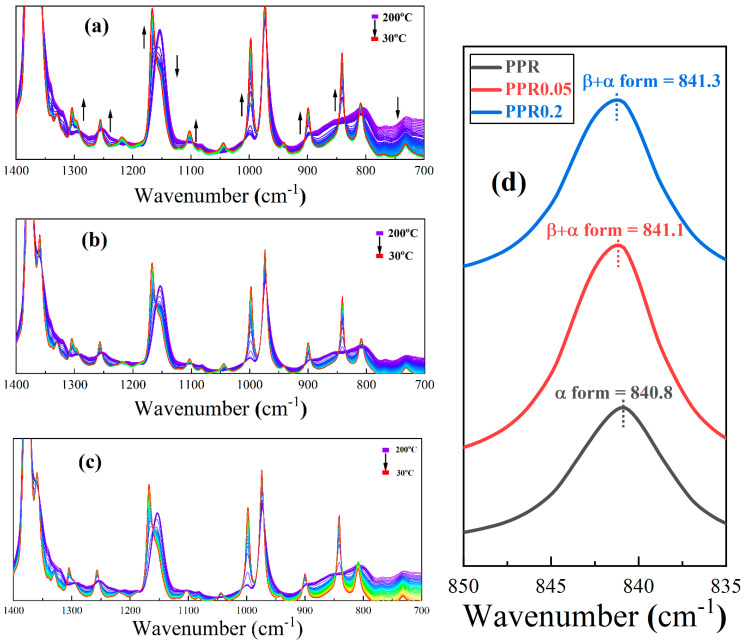
The infrared absorbance spectra of samples: (**a**) PPR0; (**b**) PPR0.05; (**c**) PPR0.2%. (**d**) The absorption peaks of the crystallized samples near 841 cm^−1^. The arrows indicate the direction of the gradual change of the corresponding crystalline bands. The gradual change from purple to red indicates that the polypropylene gradually cools and crystallizes from the molten state to the solid state.

**Figure 9 polymers-16-00827-f009:**
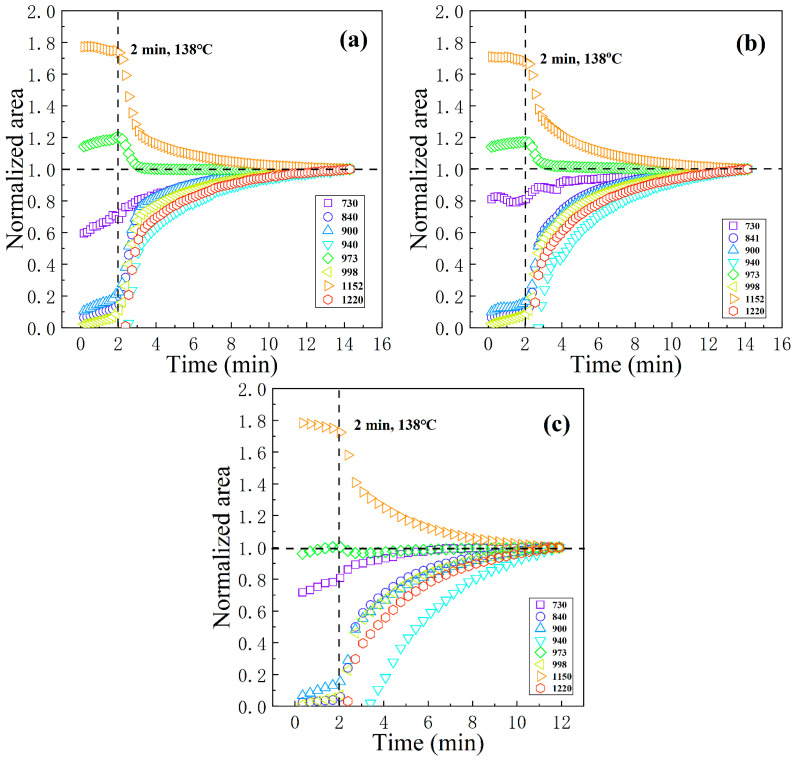
Variation of the normalized area of the characteristic absorption peak of the helical ordered length segment with cooling time in samples: (**a**) PPR0; (**b**) PPR0.05; (**c**) PPR0.2.

**Table 1 polymers-16-00827-t001:** Infrared bands of polypropylene below 1400 cm^−1^ versus the helical length [46].

Wave Number (cm^−1^)	Vibrational Mode	Corresponding Number of Monomers in Helices
808	*v*(C-C); *c*(CH)	~7
841	*v*(C-C); *r*(CH_2_); *r*(CH_3_)	12
900	*v*(C-C); *c*(CH)	~8
940	*v*(C-C); *c*(CH)	>14
973	*v*(C-C); *r*(CH_2_); *r*(CH_3_)	3–4
998	*v*(C-C); *r*(CH_2_); *r*(CH_3_)	10
1100	*v*(C-CH_3_); *r*(CH_3_); *δ*(CH)	~6
1167	*v*(C-C); *w*(CH_3_)	~6
1220	*t*(CH_2_); *δ*(CH); *v*(C-CH_3_)	14
1303	*δ*(CH)	~13
1330	*δ*(CH); *t*(CH_2_)	~13

*v*-stretching; *r*-rocking; *δ*-bending; *t*-twisting; *w*-wagging; *c*-coupled deformation.

## Data Availability

Data are contained within the article.

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
