# Peer review of "The Crystallization Morphology and Conformational Changes of Polypropylene Random Copolymer Induced by a Novel β-Nucleating Agent"

_polymers, 2024, doi:10.3390/polym16060827_

Round 1
Reviewer 1 Report
Comments and Suggestions for Authors
The authors do not specify the molecular structure/composition of the nucleating agent since they claim is under patent protection. For my experience, when a patent is deposited, a paper can be written with all the details. In the current version the experiments cannot be reproduced and it is also impossible to reason about the chemistry leading the crystallization process.
Author Response
The authors do not specify the molecular structure/composition of the nucleating agent since they claim is under patent protection. For my experience, when a patent is deposited, a paper can be written with all the details. In the current version the experiments cannot be reproduced and it is also impossible to reason about the chemistry leading the crystallization process.
Response: We thank this reviewer for the comments. We are sorry to trouble you because of our inaccurate description. In fact, we are planning to apply for a patent for this new nucleating agent, so the specific structure cannot be disclosed for the time being. The main structure of the nucleating agent is an organic complex of Ba and La with some specific ligand. It is a light rare earth nucleating agent. The structure of another rare earth nucleating agent WBG is also not specified the composition. For example, in the following article, https://doi.org/10.1002/app.29139 and https://doi.org/10.1016/j.polymertesting.2008.04.004. WBN-28 is an effective new polypropylene β nucleating agent with different structure from WBG, but it belongs to La series light rare earth. Taking into account the reviewers' comments, we have revised and redescribed the composition of the nucleating agent in the manuscript.
At present, there is no clear statement about the β nucleation mechanism or chemical action of rare earths, and we have been trying to understand it. It is only found that some structures of La series are good, and no other rare earth elements are found for the time being. We have always spared no effort to do research in the hope of obtaining more experimental data and results, just like this manuscript.
With regard to the problem of trial repetition mentioned by the reviewer, WBN-28 has begun to be popularized at present. We also describe the test method in the manuscript in detail. Using WBN-28 according to these methods can achieve a good repetition of the test. We welcome and look forward to studying and discussing with other scholars to obtain more results that are beneficial to the progress of the discipline.
Thanks again to you for the attention and helpful suggestions.

Reviewer 2 Report
Comments and Suggestions for Authors
In my opinion the manuscript “polymers-2903899” can be published in Polymers journal after a minor revision taking into account the following points.
1. In my opinion, in introduction it should be briefly described what is the chemical nature of popypropylene random copolymer (ethylene content is mentioned in Lines 94-95 but it would be better to give this information earlier). Also, please describe what are the advantages of PPR in comparison to polypropylene homopolymer. It would also be useful to briefly discuss in what ways β-crystalline polypropylene is better than its α-crystalline analogue and why is it important to find ways to obtain higher β-content samples.
2. Line 152-153. Increase of melting temperature does not necessarily mean increase of lamellar thickness. Melting temperature can also be a function defectiveness of crystallites, their stressed state, thickness of amorphous regions between the lamellae etc. Please note this in the paper text.
3. I am interested whether the high-temperature peak in DSC (Figure 1a) for the samples containing the nucleating agents reflects melting of the α-crystallites initially present in the sample or melting of α-crystallites formed in the course of sample heating after melting of β-crystallites (melting – recrystallization).
4. Figures 2-4 show POM images of different samples during their storage at 120°C at a certain time (0, 1, 2, 3 and 4 minutes of annealing). However only in Figure 2 crystallization process can be observed (nucleation in Fig. 2b, then growth in Fig. 2c,d,e). In Figures 3 and 4 crystallization process is almost over in Figure 3c-e and 4b-e) and growth of spherulites is not shown. If there was a video-recording in the course of annealing, it would be advantageous to either share these videos in supporting information or change the images in Figures 3 and 4 to (for example) taken after 30 sec, 1 minute, 1.5 minute and 2 minute of annealing so that crystallization process can be observed. I would not ask to redo these experiments if the authors do not have this data. This comment is valid only if this data is already available.
5. It is better to use term “crystallites” instead of “crystals” discussing h=behavior of semicrystalline polymers.
6. In Lines 310-312 the authors claim that β-crystalline structure can be observed in SEM images. Indeed lamellar and spherulitic structure is visible in SEM images. And indeed, from DSC and WAXS measurements we know that this spherulite is composed of β-crystallites. Was that the only logic behind this claim or the authors imply that it is possible to distinguish a spherulite composed of α-crystallites and a spherulite composed of β-crystallites based on SEM morphology only. If yes, please describe characteristic peculiarities of these different morphologies (and maybe, show SEM image of the spherulite composed of α-lamellae for comparison). Scheme of the spherulite in this figure seems to be unnesessary.
7. Is it possible to shed a little bit more light on the chemical structure of nucleating agent? Is that an inorganic complex of lanthanum?
Comments on the Quality of English LanguageThere are some sentences that need correction in terms of English. For example, in Line 149 “DSC and WAXD are frequently used to measure β-crystal”; in Line 156 “Moreover, a small amount of γ-crystal is naturally formed”, etc.
Author Response
In my opinion the manuscript “polymers-2903899” can be published in Polymers journal after a minor revision taking into account the following points.
Response: We thank this reviewer for the recognition.
- In my opinion, in introduction it should be briefly described what is the chemical nature of polypropylene random copolymer (ethylene content is mentioned in Lines 94-95 but it would be better to give this information earlier). Also, please describe what are the advantages of PPR in comparison to polypropylene homopolymer. It would also be useful to briefly discuss in what ways β-crystalline polypropylene is better than its α-crystalline analogue and why is it important to find ways to obtain higher β-content samples.
Response: PPR is an important copolymer of polypropylene. In its molecular chain, ethylene copolymerization units are randomly inserted into the propylene sequence. Since the random insertion of ethylene copolymerization unit destroyed the crystallization of propylene sequence, the thickness and crystallinity of PPR decreased significantly. Compared with PPH, PPR has higher toughness, higher ductility, moderate strength and excellent transparency, which makes it has been widely used in all walks of life.
The β-crystal of polypropylene has a great influence on the properties of products because of its unique structure. Shear yield and deformation of loose β-crystal can consume a lot of impact energy. Therefore, the material has higher toughness without too much decrease in rigidity, especially low temperature toughness. On the other hand, due to the slow mobility of the amorphous phase, the diffusion coefficient and solubility coefficient are lower, showing higher oxidation stability than α-PP. The higher content of β-crystal means that the above properties are improved greatly, and people are always trying to obtain higher content of β-crystal as much as possible.
The relevant content has been explained and modified in introduction, and new references have been introduced.
- Line 152-153. Increase of melting temperature does not necessarily mean increase of lamellar thickness. Melting temperature can also be a function defectiveness of crystallites, their stressed state, thickness of amorphous regions between the lamellae etc. Please note this in the paper text.
Response: We have deleted the inaccurate statement “indicating that the thickness of the lamella increases slightly”.
- I am interested whether the high-temperature peak in DSC (Figure 1a) for the samples containing the nucleating agents reflects melting of the α-crystallites initially present in the sample or melting of α-crystallites formed in the course of sample heating after melting of β-crystallites (melting – recrystallization).
Response: In fact, the high-temperature peak in DSC (Figure 1a) for the samples containing the nucleating agents mostly reflects melting of the α-crystallites initially present in the sample. We compared the β-crystal content calculated by DSC and XRD respectively, and the data were very close. It proved that the melting of β-crystal is rarely recrystallized into α-crystal, and the contribution to the enthalpy of α-crystal is very small.
- Figures 2-4 show POM images of different samples during their storage at 120°C at a certain time (0, 1, 2, 3 and 4 minutes of annealing). However only in Figure 2 crystallization process can be observed (nucleation in Fig. 2b, then growth in Fig. 2c,d,e). In Figures 3 and 4 crystallization process is almost over in Figure 3c-e and 4b-e) and growth of spherulites is not shown. If there was a video-recording in the course of annealing, it would be advantageous to either share these videos in supporting information or change the images in Figures 3 and 4 to (for example) taken after 30 sec, 1 minute, 1.5 minute and 2 minute of annealing so that crystallization process can be observed. I would not ask to redo these experiments if the authors do not have this data. This comment is valid only if this data is already available.
Response: You're very observant. In fact, during the experiment, we found that the crystallization rate was too fast due to the high efficiency of nucleating agent, the crystallization process is almost over in Figure 3c-e and 4b-e. Since we use the same time interval in Figure 2, we also use the same interval in Figure 3 and Figure 4 for comparison. We did not have analogous data taken after 30 sec, 1.5 minute. Your suggestion of shortening the time interval is very worthy of application to subsequent experiments. We will pay attention to this in the follow-up study.
- It is better to use term “crystallites” instead of “crystals” discussing h=behavior of semicrystalline polymers.
Response: Thanks to the reviewer's suggestion, we have made modifications and color distinctions in the appropriate places.
- In Lines 310-312 the authors claim that β-crystalline structure can be observed in SEM images. Indeed lamellar and spherulitic structure is visible in SEM images. And indeed, from DSC and WAXS measurements we know that this spherulite is composed of β-crystallites. Was that the only logic behind this claim or the authors imply that it is possible to distinguish a spherulite composed of α-crystallites and a spherulite composed of β-crystallites based on SEM morphology only. If yes, please describe characteristic peculiarities of these different morphologies (and maybe, show SEM image of the spherulite composed of α-lamellae for comparison). Scheme of the spherulite in this figure seems to be unnesessary.
Response: We have deleted the scheme of the spherulite in Figure 6. We have showed SEM image as Figure 6(a) of the α-crystalline for comparison. The corresponding descriptions in the manuscript were revised. SEM images of α-crystalline and β-crystalline can also be referred to the following paper, https://doi.org/10.1002/pcr2.10105. We have also newly included it in the reference of the manuscript.
- Is it possible to shed a little bit more light on the chemical structure of nucleating agent? Is that an inorganic complex of lanthanum?
Response: It is an organic complex of barium and lanthanum with some specific ligand. We have made modification in the manuscript. We are planning to apply for a patent for this new nucleating agent, so the specific structure cannot be disclosed for the time being.
Comments on the Quality of English Language
There are some sentences that need correction in terms of English. For example, in Line 149 “DSC and WAXD are frequently used to measure β-crystal”; in Line 156 “Moreover, a small amount of γ-crystal is naturally formed”, etc.
Response: We have modified the corresponding English expressions in the manuscript. “DSC and WAXD are frequently used to measure content of β-crystal and characterize β-crystal.” “Moreover, a small amount of γ-crystal is generated”.
Thank you again for the professional and valuable comments, we have made changes according to your comments, the changes have been highlighted, you can see whether the changes are satisfactory.

Round 2
Reviewer 1 Report
Comments and Suggestions for Authors
I think that it is not scientifically ethic and correct to publish without giving full details on all the materials used in the sample preparation, and here it is missing information of the main aspect of the paper, namely the structure of the nucleating agent. If a material is already patented , everything can be disclosed in a publication. So it is not even correct the sentence the author reported in line 96 "specific ingredients were not available due to patent protection". I am sorry but I cannot accept the manuscript in the current version.